# Numerical Studies on the Motions of Magnetically Tagged Cells Driven by a Micromagnetic Matrix

**DOI:** 10.3390/mi14122224

**Published:** 2023-12-10

**Authors:** Shanjia Tao, Jianguo Wu, Yongqing He, Feng Jiao

**Affiliations:** 1School of Mechanical Engineering, Chongqing Technology and Business University, Chongqing 400067, China; tsj1450576748@163.com; 2State Key Laboratory of Power Transmission Equipment & System Security and New Technology, Chongqing University, Chongqing 400044, China; 202011021066t@cqu.edu.cn; 3Chongqing Key Laboratory of Micro-Nano System and Intelligent Transduction, National Research Base of Intelligent Manufacturing Service, Chongqing Technology and Business University, Chongqing 400067, China; 4School of Chemical Engineering, Kunming University of Science and Technology, Kunming 650500, China

**Keywords:** magnetic, capture trajectory, capture efficiency, adhesion density

## Abstract

Precisely controlling magnetically tagged cells in a complex environment is crucial to constructing a magneto-microfluidic platform. We propose a two-dimensional model for capturing magnetic beads from non-magnetic fluids under a micromagnetic matrix. A qualitative description of the relationship between the capture trajectory and the micromagnetic matrix with an alternating polarity configuration was obtained by computing the force curve of the magnetic particles. Three stages comprise the capture process: the first, where motion is a parabolic fall in weak fields; the second, where the motion becomes unpredictable due to the competition between gravity and magnetic force; and the third, where the micromagnetic matrix finally captures cells. Since it is not always obvious how many particles are adhered to the surface, attachment density is utilized to illustrate how the quantity of particles influences the capture path. The longitudinal magnetic load is calculated to measure the acquisition efficiency. The optimal adhesion density is 13%, and the maximum adhesion density is 18%. It has been demonstrated that a magnetic ring model with 100% adhesion density can impede the capture process. The results offer a theoretical foundation for enhancing the effectiveness of rare cell capture in practical applications.

## 1. Introduction

The magnetic capture technique is frequently employed because of its outstanding integration and controllability [1,2,3,4,5,6]. It has been demonstrated that the prudent use of external magnetic fields to effectively capture circulating biomarkers, such as proteins [7], nucleic acids [8,9], and cells [10,11,12,13,14], is a useful technique for disease detection and treatment. It is very important to have both a magnet matrix with quick magnetic response properties and microchannels for controlling the movement, separation, and extraction of compounds at the cellular and even molecular levels. Accurately controlling magnetic particles has served as the basis for constructing a magneto-microfluidic platform. Nano-magnetic particles are widely used in bioengineering [15,16,17,18], chemical pharmacy [19], clinical medicine [20,21,22,23,24], and other fields due to their uniform particle size and fast magnetic reaction. Especially, peripheral-blood-circulating tumor cells, or CTCs, are a special type of cell associated with a higher chance of survival for cancer patients [25]. Isolating and enriching this type of rare cell is essential for creating a therapeutic platform because of its extremely low blood concentration [26,27,28].

Several investigations have been carried out on magneto-microfluidic platforms integrated with external magnetic fields to separate magnetic particles in fluids. Chen et al. used submicron-scale magnetic structures to produce strong local-magnetic-field gradients in a channel, and many distributed trapping spots were formed [29]. Xu et al. developed a three-dimensional inverse opal photonic crystal substrate with high cell affinity and capture efficiency [30]. Simultaneous simulations were also carried out to assess the feasibility of magneto-microfluidic platforms. Jake et al. proposed a magnetic model with uniaxial symmetry [31]. The absolute maximum torque formula is derived from this continuous model, which unifies the anisotropic magnetic models. Kazunori et al. provided an analytical model with magnetic force and gravity as the driving forces based on the consideration of the mechanical properties of the medium [32]. Kim et al. suggested a bionic 3D model of cilia with the magnetic-field intensity, object length, external magnetic-field angle, and accurate angle as independent variables [33]. Mahla et al. tracked the dynamic characteristics of circulating tumor cells using the MagRC method. Subsequently, they observed a three-dimensional capture curve, as demonstrated by fluorescence immunoassay [34]. But a detailed examination of the cell’s path, taking the labeled magnetic beads into account, was necessary.

A two-dimensional numerical model is established in this paper to forecast the motions of magnetically tagged cells controlled by a micromagnetic matrix. The micromagnetic matrix is located on the bottom of a microchannel, where the labeled cells are transported into the magnetic-field area. We divide the channel into a magnetic-rich region and a sparsely magnetic area to better describe the capture process. The labeled cell trajectories can be obtained after solving the coupled Navier–Stokes and Maxwell’s equations. Since the number of magnetic beads adhering to the cell surface is unpredictable, we introduced adhesion density to evaluate the impact of the magnetic properties. The optimal and maximum adhesion densities and the trapping trajectories of labeled cells are investigated. When the attachment density approaches 100%, we use a magnetic ring model to discuss this extreme case.

## 2. Physical Modelling

### 2.1. Constitutive Equations

Every magnetic particle in the microfluidic channel is subject to three different forces, each depicted in Figure 1a and Figure 2a with corresponding directions. We compute it in the following manner, considering that the pipe’s flow field is a typical low Reynolds number viscous incompressible flow with a sluggish flow rate. The fluids are incompressible, meaning they exist as *ρ_f_* = *const*, *∂r*/*∂t* = 0. Moreover, the *N-S* equation is simplified as follows:(1)$$\frac{\partial u_{f}}{\partial t}=-\frac{1}{\rho_{f}}\nabla p+F_{T}+\frac{1}{\rho_{f}}\nabla \cdot \tau$$

***u****_f_* stands for fluid velocity; *ρ_f_* for fluid density; *p* for pressure; ***F****_T_* denotes the fluid’s load on the magnetic particles, and the sum of pressure and viscous force; and *τ* is viscous force. There is an equation for when the viscosity coefficient is constant or if there is no temperature change:(2)$$\frac{1}{\rho_{f}}\nabla \cdot \tau =v∆u_{f}$$
where *v* denotes a scalar velocity. The *N-S* equation for incompressible flow is obtained by substituting Equation (2) into Equation (1):(3)$$\rho_{f}\frac{\partial u_{f}}{\partial t}=-\nabla p+\rho_{f}F_{T}+\mu ∆u_{f}$$

The *N-S* equation is used to calculate the flow-field distribution:(4)$$\left\{\begin{array}{l} \rho_{f}\left(u_{f}\cdot \nabla \right)u_{f}=\nabla \cdot \left[-pI+\mu \left(u_{f}+{\left(u_{f}\right)}^{T}\right)\right]\\ \rho_{f}\nabla \cdot u_{f}=0 \end{array}\right.$$
where *μ* is the hydrodynamic viscosity coefficient, and ***I*** is the unit diagonal matrix. The boundary condition at the exit is as follows, since the boundary condition at the entry is set to ***u****_f_* = −***n****U*_0_ (where ***n*** is the normal vector and *U*_0_ represents the initial velocity scalar of the flow field):(5)$$\left[-\rho_{f}I+\mu \left(u_{f}+{\left(u_{f}\right)}^{T}\right)\right]n=0$$

The bottom heterogeneous magnetic block, which causes an uneven distribution of magnetic-field lines, affects the flow field along the entire channel, and the spatial magnetic-field strength meets the following equation:(6)∇B=0

In the equation, ***B*** stands for magnetic-flux density. The Maxwell stress tensor is introduced:(7)$$T_{e}=\varepsilon_{0}(E\cdot E-\frac{1}{2}E^{2}\delta )+\frac{1}{\mu_{0}}(B\cdot B-\frac{1}{2}B^{2}\delta )$$
where ***E*** is the electric-field strength, *δ* is for the Kronecker symbol, and *μ*_0_ is for vacuum permeability. Considering this is a pure-magnetic-field system, insert ***E*** = 0 into
(8)$$F_{m}=\nabla \cdot T_{e}=\frac{1}{\mu_{0}}\left[\left(\nabla \cdot B\right)B+\left(B\cdot \nabla \right)B\right]-\frac{1}{2\mu_{0}}\nabla B^{2}$$

Fluid loads are also imposed on solids in channels, and the fluid–solid coupling equation can be determined utilizing the Navier–Stokes equation as follows:(9)$$F_{T}=-n\cdot \left(-\rho_{f}I+\mu \left(\nabla u_{fs}+{\left(\nabla u_{fs}\right)}^{T}\right)\right)$$

In the formula, ***u****_fs_* is the speed at which the flow field and the solid meet. Gravity cannot be disregarded when examining its track because the density of magnetic particles is significantly higher than that of fluid, that is
(10)$$F_{g}=m_{s}g\frac{\rho_{s}-\rho_{f}}{\rho_{s}}$$

The following formula can be used to determine how the rigid-body model will move:(11)$$\rho_{s}\frac{\partial^{2}u_{s}}{\partial t^{2}}=F$$
where ***u****_s_* is the solid velocity, *ρ_s_* is the solid density, and ***F*** is the external force. The total of gravity is ***F****_g_*, fluid load is ***F****_T_*, and magnetic load ***F****_m_* is ***F***.

### 2.2. Numerical Algorithm

The commercial software COMSOL 6.0 Multi-Physics was used to run a two-dimensional (2D) computational fluid dynamics (CFD) simulation of the micromagnetic channel, and the non-ideal motion behavior produced by the interaction of the flow field and the magnetic field was investigated. The anti-slip condition of viscous fluid in fluid dynamics determines that the fluid at the solid boundary has zero velocity relative to the border, and the simulation model meets the Dirichlet boundary condition, as shown in Figure 1a. The fluid–structure interaction domain is a freely moving deformable mesh on which the Navier–Stokes equations are solved. Yeoh smoothing handles mesh deformations associated with the original domain shape.

Figure 1 depicts the schematic representation of the capture mechanism and the adhesion of magnetic particles.

**Figure 1 micromachines-14-02224-f001:**
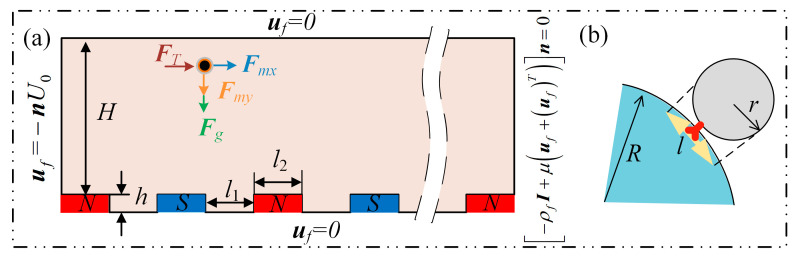
Model schematic: (**a**) Schematic diagram of the device; (**b**) magnetic particles adsorbed by cancer cells.

This study employs finite element and dynamic mesh methods to solve partial differential equations. The magnetic load ***F****_m_* of the target is first computed by calculating the magnetic-field distribution along the whole magnetic channel using the Gaussian magnetic law. Utilizing the *N-S* equation, we determine the flow-field distribution in the pipeline and, using that information, determine the flow-field thrust ***F****_T_* on the target. We calculate the overall force of the solid by adding a constant for gravity, and then we use the equilibrium differential equation to determine the solid’s velocity ***u****_s_* and displacement Δ*S* throughout Δ*t* time while an external force is acting on it. The grid then begins to self-check, meaning that the curvature factor must be less than 0.2, and the smallest element cannot be more minor than 0.002 mm. The magnetic-field flow field can be computed once more if the precision requirements are satisfied; if not, the update node is re-divided to satisfy the accuracy requirements. The process is repeated until the solid material is adsorbed at the bottom of the magnetic channel. In the Figure 2, the precise mathematical procedure is displayed.

**Figure 2 micromachines-14-02224-f002:**
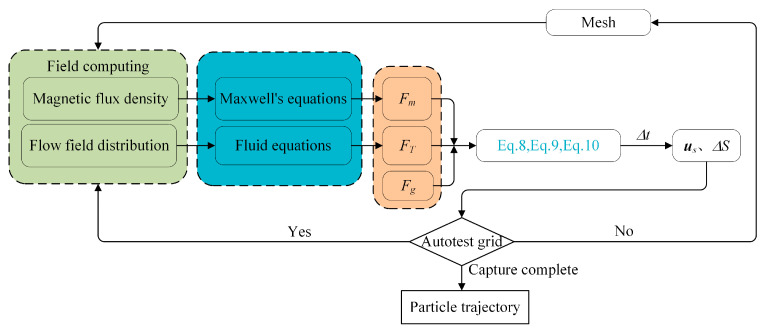
Target-acquisition trajectory-calculation flow chart.

The magnetic particle density is set to 4 × 10^3^ kg/m^3^, the cell density and fluid are set to 1.05 × 10^3^ kg/m^3^, the relative magnetic permeability is set to 1, the magnetic permeability is set to 200, *U*_0_ is set to 100 μm/s, and the magnetic potential of the magnetic block is set to ±0.5 A, respectively. The reference [35] is mainly cited while setting the geometric parameters in this paper. The parameters of magnetic particles are referred to in the references [1,25]. Table 1 below lists all geometric parameters in the computational domain of this paper:

Several calculation domains are segregated with varying degrees of precision to ensure the accuracy of the calculation results. Grids are created within the built-in computation domain after the parameter setting is complete. The magnetic particles, the target interface, and the magnetic block interface are all encrypted to varying degrees. The minimum angle is preserved above 70 degrees, and the initial grid quality is maintained above 0.6. Figure 3 displays the original grid division:

## 3. Numerical Results and Discussion

### 3.1. Magnetic-Field Analysis

The magnetic load, which acts as a trapping force, is a significant factor impacting the trapping outcome, according to the equilibrium differential equation, demonstrating that external force plays a vital role in the movement of magnetic particles. Equation (8) shows that, while the magnetic permeability of the magnetic particles is constant, the magnetic load’s magnitude mostly depends on the rate at which the magnetic field changes. This change rate varies significantly depending on where one is in the magnetic channel.

According to the direction of the magnetic-field lines, the magnetic field in the magnetically rich region is categorized into four groups in this paper: the type I magnetic field is directly above the NS magnetic block; the type II magnetic field is above the space between adjacent magnetic blocks; the type III magnetic field is above the intersection of the magnetic block, and the gap; and the type IV magnetic field is situated at the corner of the magnetic block and the space. Figure 4a demonstrates the distinct directions of the four partitions’ magnetic-field lines. The magnetic-load lines of the type I magnetic field are nearly parallel to the surface of the magnetic block and are evenly distributed; the type II magnetic field is roughly parallel to the channel; the type III magnetic field is at an angle with the bottom of the channel at almost 45 degrees; the type IV magnetic-load lines are located close to the vertex of the magnetic block, where the magnetic field is abnormally concentrated.

Since the magnetic particles are tiny and homogenous, their magnetization can be effectively independent of an external magnetic-field source. Equation (8) and reference [29] show that while the magnetic permeability of magnetic particles is constant, the amount of magnetic force experienced by the magnetic particles is mainly determined by the rate of change of the magnetic field. As a result, more emphasis is given to how magnetic load changes in the magnetically dense region. Two characteristics of magnetic load are evident in Figure 4b: (1) magnetic load nearly exclusively resides on the surface of magnetic particles; (2) magnetic load is concentrated at the vertical incidence and exit of the magnetic-load lines. Ohm’s law of magnetic circuitry and the connection requirement of a constant magnetic field can be used to explain this phenomenon. As shown in Figure 4c, the magnetic-field line originates from the magnetic block of the N pole and travels through air, magnetic particles, and air before returning to the magnetic block of the S pole.

This procedure can be compared to a magnetic circuit in which three reluctances are connected in series with the magnetomotive force, as depicted in Figure 4d. The following describes Ohm’s law for a magnetic circuit:(12)$$\Phi =\frac{F_{A}}{R_{m}}$$

In the equation, *Φ* stands for the magnitude of the magnetic flux moving through the magnetic circuit, *F_A_* for the magnetomotive force (also known as the magnetic voltage drop), and *R_m_* for reluctance (its magnitude is inversely proportional to the magnetic permeability and proportionate to the length of the material). The magnetic particles’ magnetic permeability is 200 times more than the vacuum, and their diameter is significantly less than the air gap’s length. The magnetic pressure drop on any magnetic circuit can be disregarded to guarantee that the magnetic flux passing through the magnetic circuit is of the same magnitude, so there is no possibility of the formation of a gradient magnetic field inside the magnetic particle, making it impossible to produce the magnetic load inside.

The magnetic-permeability parameter describes the process of magnetizing magnetic particles using an external magnetic-field source. The ratio of the magnetic-induction intensity to the magnetic-field intensity when the magnetic medium is magnetized is known as magnetic susceptibility, and it is denoted using the symbol *μ*. It exists as
(13)$$\mu =\frac{B}{H}$$

*μ*_0_ is the vacuum magnetic permeability, and *μ_r_* is the relative magnetic permeability. It exists as
(14)$$\mu =\mu_{0}\mu_{r}$$

Since *μ_r_* is typically a constant, the parameter for relative permeability is typically used to describe how easily a material is magnetized. Moving through the micromagnetic channel has little impact on the overall magnetic field because of the magnetic particles’ tiny size. Still, because the magnetic permeability of the particles is hundreds of times greater than that of the fluid, the magnetic field is significantly distorted (Figure 4b). The connection formula for a constant magnetic field can be used to determine how much distortion there is:(15)$$\frac{\tan\alpha_{1}}{\tan\alpha_{1}}=\frac{\mu_{r1}}{\mu_{r2}}$$

*α*_1_ and *α*_2_ in the formula above stand for the incident and exit angle of the magnetic-load line interface, respectively; *μ_r_*_1_ and *μ_r_*_2_ stand for the fluid’s and magnetic particles’ respective relative magnetic permeabilities. The above formula shows that when the magnetic-field lines are injected into the magnetic particles from the liquid, they will be bent to the center of the magnetic particles. As a result, the area where the magnetic-field lines are perpendicular to the incident magnetic particles is where the magnetic-field lines are enriched. Minimal magnetic-field lines are present on both sides, away from this area. As a result, as illustrated in Figure 4b, the magnetic load is primarily concentrated in the incident and outgoing zones.

### 3.2. Single-Magnetic-Particle Computing

The magnetic-load ratio is put forth to take the influence of units out of the calculation results. The load on the particle surface is dimensionless based on the magnetic load on the type III region (*F*_3_ in the figure refers to the magnetic load on the type III region). The simulation calculation is limited to the force state of the particle’s exterior in four typical fields because there is no force inside it. The ratio of magnetic load experienced by magnetic particles in the horizontal and vertical directions under the four different magnetic fields is plotted in Figure 5. The figure shows that the magnetic load on the single magnetic particle differs noticeably depending on the four types of magnetic fields.

The magnetic particles’ transverse- and longitudinal-magnetic-load ratios in the type I magnetic field are 0.0074 and −3.6, respectively (|*F_mx_*| << |*F_my_*|). From this, it can be shown that the transverse magnetic load barely affects the magnetic particles in the type I magnetic field, being only vulnerable to magnetic loads along the longitudinal axis. The magnetic particles’ transverse- and longitudinal-magnetic-load ratios (|*F_mx_*| >> |*F_my_*|) are 1.8 and −0.0056, respectively, in type II magnetic field. The magnetic particles in the type II magnetic field almost entirely lack longitudinal magnetic load and only exhibit a transverse magnetic load, in contrast to type I magnetic-field features. The transverse and longitudinal magnetic loads of the magnetic particles in the type III magnetic field are at the same amount because the transverse- and longitudinal-magnetic-load ratios of the magnetic particles in the type III magnetic field are 2.5 and −2.7, respectively (|*F_mx_*| ≈ |*F_my_*|). The magnetic particles’ transverse and longitudinal magnetic-load ratios are 79.0 and −164.6, respectively, in the type IV magnetic field. As observed, this region’s magnetic particles have a magnetic load that is significantly higher than those of the other three types of magnetic fields.

The single magnetic particle is positioned at a certain distance from the channel’s bottom, and the mathematical model indicated above determines the magnetic particles’ course. The geometric parameters of the channel are dimensionally processed based on the size of the magnet block, and the height (length) ratio is defined as the ratio of the height (length) of the research object from the magnet block to the height (length) of the magnet block. (The rest of the captured trajectoriesin the text uses the same coordinate-processing method as the process described above.) The computation’s outcomes are displayed in Figure 6:

The motion trajectory map of a single particle under gravity and without gravity is calculated in Figure 6. It is discovered by contrasting the two that their motion trajectories are noticeably dissimilar. The magnetic particle falls more quickly under the effect of gravity. Due to the great density of magnetic particles, it is significantly larger than the density of the surrounding liquid; magnetic particles fall into the fluid too quickly to allow for a detailed study of their motion trajectories under the impact of gravity, magnetism, and flow fields. Still, gravity is constant, so the research is focused on how magnetic particles move when there is no gravitational pull on them. The importance of magnetic and flow fields in attracting magnetic particles is highlighted by the ignorance of the influence of gravity. The magnetic particle’s route can be observed in the image to have distinct stages, and the final section of the trajectory exhibits a vertical downhill tendency that is wholly incompatible with the initial trajectory. Therefore, the gravity-free track is divided into the driving and capture stages. Not all microfluidic systems are propelled by gravity. A microfluidic system called the Mag-Gradient Chip was proposed by Bongseop et al. [26]. that solely takes into account the fluid resistance and magnetic force of the medium that the particles are affecting. This magnetic-gradient-based microfluidic system ignores gravity.

The falling of the magnetic particles occurs periodically during the driving phase. If the magnetic particle exhibits a downward tendency in one region, it will stay vertically stationary in the subsequent region (the quiescent zone) before falling again (the drop zone). Moreover, the quiescent zone is typically found above the air gap, whereas the drop zone is generally located above the magnetic block. The magnetic particle comes closer and closer to the magnetic block as the capture phase continues, and the outer region of the quiescent zone progressively changes into a drop zone. Examining the causes of this occurrence from the standpoint of load, early in the driving stage, only type I and type II space magnetic fields exist, the magnetic lines of force are relatively scarce, and the distance between the magnetic particle and the magnetic block is quite far; in the type I magnetic field, the magnetic particle exhibit a slight falling motion; in the type II magnetic field, they are longitudinally stationary. A type III magnetic field starts to form at the intersection of type I and type II space magnetic fields as the fall progresses and the magnetic particles eventually enter the magnetic-rich zone; the type III magnetic field will apply downward ***F****_m_* to the magnetic particles, causing the descending area to widen progressively and the static site to contract gradually. The stage of trapping takes place later in the magnetic-particle movement. The magnetic particle appears to be moving nearly vertically at this point. The magnetic particle has moved into the zone of the type IV magnetic field, which is the cause of this event. The magnetic particle falls quickly in this region because ***F****_m_* affects it much more than in the other three areas.

### 3.3. Multi–Magnetic-Particle Computing

#### 3.3.1. Force Analysis

The grid division and parameter settings for the cell capture computation are presented in Section 2.2. The previously discussed mathematical model can still be employed in cell capture computation. The changing magnetic load during the movement is still the focus of the force analysis. A contact pair calculation model based on the penalty function was introduced to calculate the trajectory of the marked cells before the collision, using the example of the case of six MPs attached to establish a universal simulation model for the micromagnetic matrix capturing marked cells. The research goal is a rigid structure to ensure that the MPs attached to the surface remain in the exact relative location throughout the capture procedure. The size of the cell and the size of the magnetic block are in the same order of magnitude in the model established in this paper, so the MPs on the same cell must be in different types of magnetic fields after entering the magnetic-rich region, so they are subjected to magnetic loads of different sizes and directions. Then, rotation occurs, as shown in Figure 7.

Laminar flow and magnetic fields work together to cause the tagged cells to rotate. The fluid’s flow velocity is lowest close to the wall and at a maximum in the channel’s center. The tagged cells rotate due to the viscous force and pressure difference they encounter in the fluid. Figure 7 uses a cell with six magnetic particles attached as an example to examine how a cell is captured. According to the previously mentioned space magnetic-field-division rules, when the cell is in the magnetic-lean region at the height of the channel, the force of the magnetic field on the magnetic particles is relatively small and negligible. As the fall progresses, the target cell gradually enters the magnetic-rich region, and the six magnetic particles are in various types of space magnetic fields, as shown in Figure 7a. The type I space magnetic field contains the magnetic particles three and four, whereas the remaining particles are randomly distributed throughout the sparse magnetic region. Most of the magnetic particles enter the magnetic-lean region as the fall and rotation advance (as can be seen in Figure 7b; the four magnetic particles two, three, four, and five are all in the type III space magnetic field), which causes the rotation to speed up. The magnetic load encourages the target object to produce a rotational torque as it falls, which causes the target object to rotate. Experimental research by Mahla et al. revealed that cells moving through the capture zone will be deflected and captured, and they also confirmed that the magnetic force acting on the magnetic particles is greatest at the micromagnet’s edge [34]. They did not, however, investigate how this rotation impacts capture progress. We looked into the cause of this rotational effect from a force standpoint in our study.

The target cell’s rotation during the movement will also be influenced by gravity and the force of the flow field. The rotation angle of the target item at various times is determined by utilizing the rotation mechanics formula using the rotation torque and the moment of inertia to calculate the cell’s rotation state (angle) during the entire procedure. Rotational torque ***M*** = *r* × ***F***, where *r* is the distance between the force’s point of action and the target’s center, and ***F*** is the force itself. The magnitude of the rotating torque can be described as follows, where ***F*** is the sum of the three external forces in this paper:(16)$$M=\iint_{S+\sum_{i=1}^{n}S_{i}}r\times F_{g}ds+\int_{l}r\times F_{m}dl+\int_{L}r\times F_{T}dl$$

In the formula, ***S*** stands for the area containing the target object, ***S****_i_* for the site, including the *i*-th magnetic particle, *l* for the magnetic particle’s contour line, and *L* for the outer contour line of the target object. The following equation can be used to determine the target’s moment of inertia:(17)$$I=\iint_{S}r^{2}\rho ds=\iint_{S_{0}}r^{2}\rho_{0}ds+\sum_{i=1}^{n}\iint_{S_{i}}r^{2}\rho_{1}ds$$

The rotational angle at various moments can be calculated using the rotational mechanic’s equation ***M*** = ***I***(d^2^*α*/d*t*^2^). This work introduces a new variable, the attachment density *η*, whose definition is as follows, to represent the number of magnetic particles on the target conveniently:(18)$$\eta =\frac{\sum_{i=1}^{n}l_{i}}{L}$$

In the equation, *i* stands for the number of magnetic particles present in the target, *l_i_* for the effective length of the magnetic particles affixed to the target’s surface, and *L* for the target’s circumference. The target cell’s rotation angle at various moments with and without the magnetic load is calculated using the attachment density of 13% (attached with six magnetic particles) as an example, as illustrated in Figure 7c. Because the flow velocity in the middle area of the pipe flow is more significant than that in the two side sections, we can see in the image that the cell rotates under the influence of magnetic load and rises first before falling. The target cell is initially located above the channel, the flow velocity below is greater than above, and the rotation is counterclockwise. The process ends when the target cell reaches the middle of the pipe and the flow velocity at the top and bottom are equal. If it keeps falling, the target reliably rotates clockwise when the higher flow velocity exceeds the lower flow velocity. The earlier hypothesis that the magnetic load is negligibly weak in the magnetic-lean zone and may be ignored is confirmed by the practically exact intersection of the two curves in the first part; however, the curve diverges in the second half. As the initial rotation tendency of the ***F****_m_* ≠ 0 curve is enhanced, the curve of ***F****_m_* = 0 still follows the prior rotation trend. The spinning tendency is strengthened while moving from a zone of magnetic leanness into a region of magnetic richness due to the imbalanced magnetic load acting on the magnetic particles.

#### 3.3.2. Motion Analysis

This paper estimated the capture trajectories of numerous groups of targets with various attachment densities and compared them, as shown in Figuer 8a, to examine the impact of attachment density on the capture target. The capture process of various attachment densities can be loosely separated into three stages, as can be seen by looking at the capture curve:1.Initial phase:

This phase occurs at the start of the movement when the target cell is in the magnetic-lean region and is less impacted by the magnetic load. At this point, the gravity of the target cell is balanced with the thrust of the flow field, causing the cells to descend in the pipeline at a constant speed.

2.Competition phase:

The target’s magnetic particles progressively enter the magnetically rich zone as the fall continues, subject to the imbalanced magnetic load. The ensuing rotating torque will hasten the target’s initial tendency to rotate, increasing its propensity to a specific magnetic block. The key to this procedure is that the magnetic blocks are vying with one another to form the strongest magnetic field possible, which will attract the target. It is also this competition that leads to uncertainty in the trajectories during capture.

3.Capture phase:

Indeed, the competition phase’s outcomes are embodied in the capture phase. The magnetic field of the magnetic block at the bottom of the channel gradually captures the magnetic particles at the bottom of the cell as the competition stage advances. The target moves almost vertically before becoming adsorbed on a specific magnetic block. The magnetic block’s left side is more likely than its right side to absorb the target when subjected to the force of the flow field. The height of the target object from the magnet matrix can be used to distinguish between three stages: the initial phase occurs when the distance from the magnetic block is more significant than 4.8 times the height; the competition stage occurs when the space is between 3.5 and 4.8 times the height; and the capture phase occurs when it is within 3.5 times the height of the magnetic block. Figure 8a shows that when the attachment density rises, the capture point moves farther from the starting position, and the initial phase progresses more quickly. However, the slope at the end of the capture curve does not demonstrate a pattern of significant rise, indicating that the rate of decline in the capture phase does not increase proportionally with the increase in attachment density.

Figure 8b describes the change curve of the magnetic-charge ratio of the research object before capture under different attachment densities. (Dimensionless processing is performed on the magnetic force, and the processing method is the same as in Figure 5). As can be seen, when the attachment density falls within interval I, there is a nearly linear relationship between the two, and the falling speed rises in direct proportion to the attachment density. As the attachment density is increased to the range of interval II, the two continue to show a linear rise connection, but the magnetic-load ratio does not increase considerably, implying that the benefits of growing attachment density in this area are much smaller than those of the increasing interval I. When the attachment density rises once more and reaches break III, the decrease in capture speed occurs instead. The optimal economic attachment density is 13% of the median value of intervals I and II. The maximum attachment density is 18% of the median value of intervals II and III. The cost of the experiment rises with the number of particles adsorbed on the labeled cell surface since they are harder to catch, need a longer microchannel, and are farther away from the capture point. It is possible to successfully capture the target at a fixed point and create an excellent downstream capture interval by managing the attachment density within a reasonable range. The two attachment densities provide crucial production guidance. On the assumption that the goal capture efficiency is kept constant, it is found that appropriately lowering the number of magnetic particles attached to the cells can reduce production costs and operational challenges.

### 3.4. Extreme Case

Figure 8b demonstrates that the magnetic-force ratio falls rather than increases as the adhesion density rises above a specific threshold. To analyze why the capture efficiency becomes slower (capture saturation) when the attachment density exceeds the maximum attachment density, this paper calculates an extreme attachment density, that is a 100% attachment density. It wraps the cell’s perimeter in a magnetic ring made of magnetic particles to study why the capture efficiency slows down (capture saturation) when the attachment density of magnetic particles exceeds the maximum attachment density. In Figure 9, the motion trajectory is displayed. The figure shows how the cell’s trajectory, which rises upward after dropping a certain distance, deviates from conventional cognition. As the cell has not yet contacted the magnetic block at the bottom, neither gravity nor the thrust of the flow field will pull the target upward during the duration of the capture operation. The rising period of 0.652 s is chosen as the reference point, and Maxwell’s equations examine the target’s magnetic-load ratio. In Figure 10b, the magnetic-load ratio chart is displayed.

The magnetic field near its location will be more distorted by a magnetic ring structure with a specific spatial size than by a single magnetic particle, according to Figure 10a. The magnetic ring’s magnetic-field lines all opt to pass via the region with the least magnetic resistance. The magnetic-load lines incident at various places and are either converged or shunted until they depart the magnetic ring because the entire magnetic ring is in a closed mode. From the A area, the above magnetic-load line enters the magnetic coil and divides into the left and right directions. The magnetic-load lines that incident on the left and right sides of the A region are all pointing downward. The magnetic-load line density is at its maximum in the B and D areas, and it finally converges and emits the magnetic ring in the C area. Figure 10b illustrates the longitudinal magnetic load *F_my_* at the magnetic ring’s outer and inner interfaces in this study. The magnetic load on the inner and outer rings of the magnetic coil is symmetrical on the *Y* = 0 axis, as seen from the figure. The magnetic ring’s magnetic load has four peaks, the biggest of which may be seen in the region between A and B. This is the case because the magnetic-field strength in the B region is the strongest, and the maximum magnetic-field gradient must exist between the A region with the lowest magnetic-field strength and the B region with the highest magnetic-field strength. The magnetic-load curves at the four locations must be at the extreme points, and there must be a peak between any two adjacent areas. The magnetic load on the magnetic ring is around 7 × 10^−4^ N × m^−3^, and the gravity on the magnetic coil is approximately 1.45 × 10^−5^ N × m^−3^, according to the integration of the longitudinal magnetic loads of the inner and outer rings. The cell must rise upward since the magnetic load is about 50 times stronger than the gravity load, which lowers the capture efficiency.

### 3.5. Documentary Support

The primary reference for choosing the channel parameters is [34], and the primary reference for choosing the magnet matrix’s magnetic potential is [35].

The reference [29] and reference [36] were examined in this paper to ensure that the magnetic-field analysis was accurate. The *B_x_*, *B_y_*, and ∇*B_x_*, ∇*B_y_* curves above the magnetic block were calculated in the literature [29] by determining the magnetic-field gradient of a single micro-magnet element. The magnetic-flux density contour curves at the top and bottom of the magnetic channel exhibit periodic oscillation features, typically rising first and then decreasing, and the bottom curve has a higher magnetic-flux density than the top. The traits align with the ones shown by the computation findings in this research.

The flow-field calculation is relatively easy, and reference [17] can be used to confirm it. The cloud picture of the flow-field distribution in Figure 2d matches the characteristics of the cloud image calculated in this research. The reference [17] employed finite element (COMSOL 5.1 Multiphysics) software to numerically calculate the velocity field of the microfluidic microporous device. The schematic illustration of the device shown in Figure 10 in [31] is consistent with the capture model provided in this article, which depicts the channel as having a bumpy structure.

The research conclusions from [34] were contrasted with those from this study to ensure accurate simulation results. The capture curve (3D) described in the literature [34] moves uniformly and reasonably smoothly in the first half. However, the motion becomes violent when the particle approaches the magnetic block. This characteristic is in line with the findings of the calculations presented in this study, demonstrating the model’s strong universality.

## 4. Conclusions

This study builds a basic model that relies on capturing individual magnetic particles and labeled cells within a channel. It has been verified through force and moment computations that the magnetic force only acts on the exterior of labeled cells. The resulting rotation of cells changes drastically when the cell is transported from the magnetically sparse region to the rich area. Three stages comprise the capture process: the first, where motion is a parabolic fall in weak fields; the second, where the motion becomes unpredictable due to the competition between gravity and magnetic force; and the third, where the micromagnetic matrix finally captures cells.

The cell surface was exposed to magnetic forces in multiple directions, leading to irregular movement trajectories. Analysis of the capture trajectories of labeled cells with varying attachment densities yielded optimal and maximal attachment densities. The calculated attachment densities are 13% and 18%, respectively. If the adhesion density continues to increase after exceeding the optimal value, the increase in capture efficiency will slow down. The capture efficiency will decrease if the adhesion density grows after exceeding the maximum value. As the adhesion density becomes nearly 100%, the magnetic load on the inner and outer rings is exceptionally symmetrical. The target cells migrate upward due to the calculated magnetic load, which has an upward direction at certain points and is much greater than gravity. This makes it difficult for the magnet matrix to capture the target cells. As a result, the capture efficiency decreases.

## Figures and Tables

**Figure 3 micromachines-14-02224-f003:**
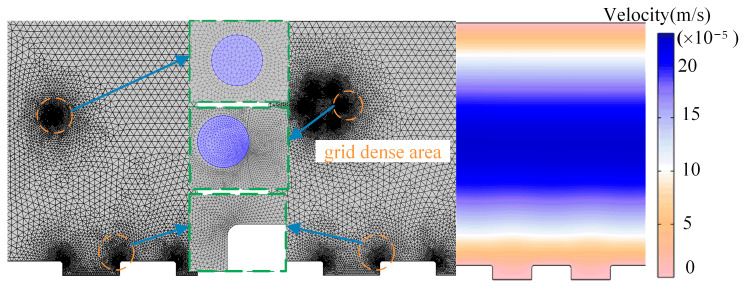
Computing domain grid-division diagram.

**Figure 4 micromachines-14-02224-f004:**
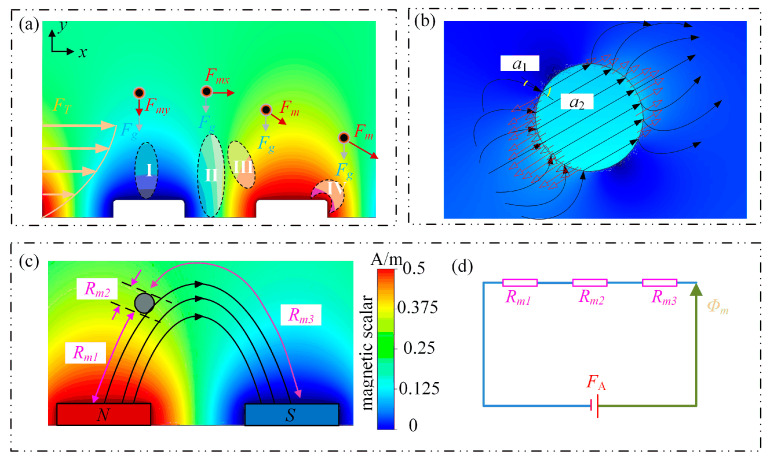
Force analysis of magnetic particles: (**a**) Schematic diagram of the division of the magnetic field in the magnetic-rich region; (**b**) schematic diagram of the force on magnetic particles; (**c**) schematic diagram of the equivalent magnetic circuit of magnetic particles; (**d**) circuit diagram.

**Figure 5 micromachines-14-02224-f005:**
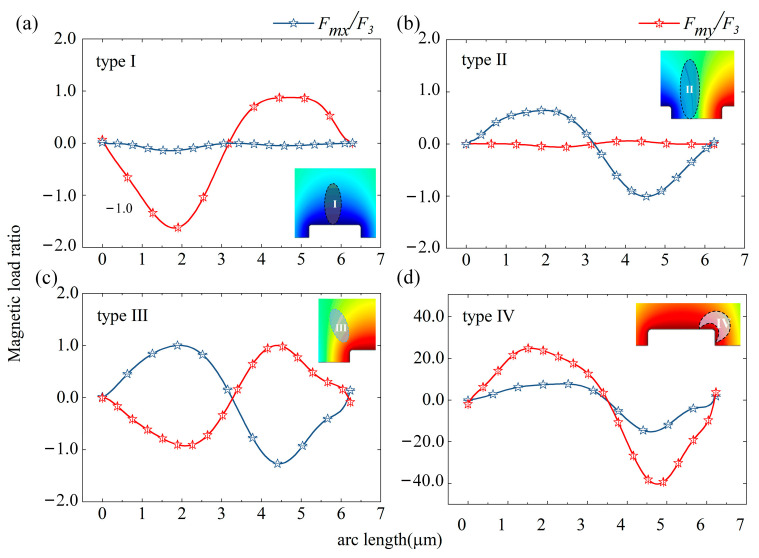
Surface-force curve of magnetic particles in four types of magnetic fields: (**a**) type I; (**b**) type II; (**c**) type III; (**d**) type IV.

**Figure 6 micromachines-14-02224-f006:**
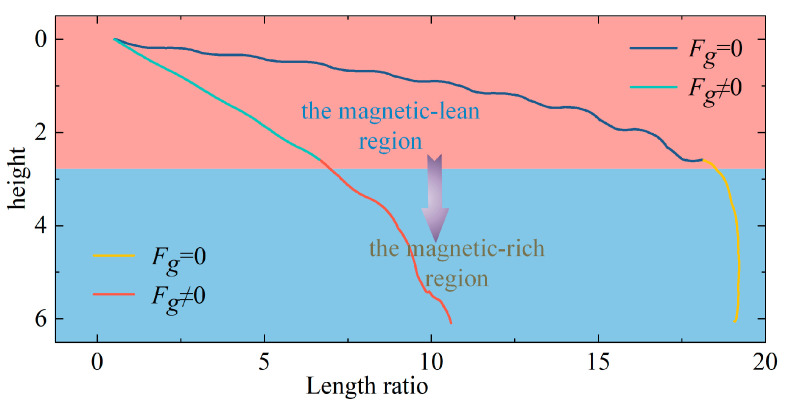
Magnetic-particle trajectories.

**Figure 7 micromachines-14-02224-f007:**
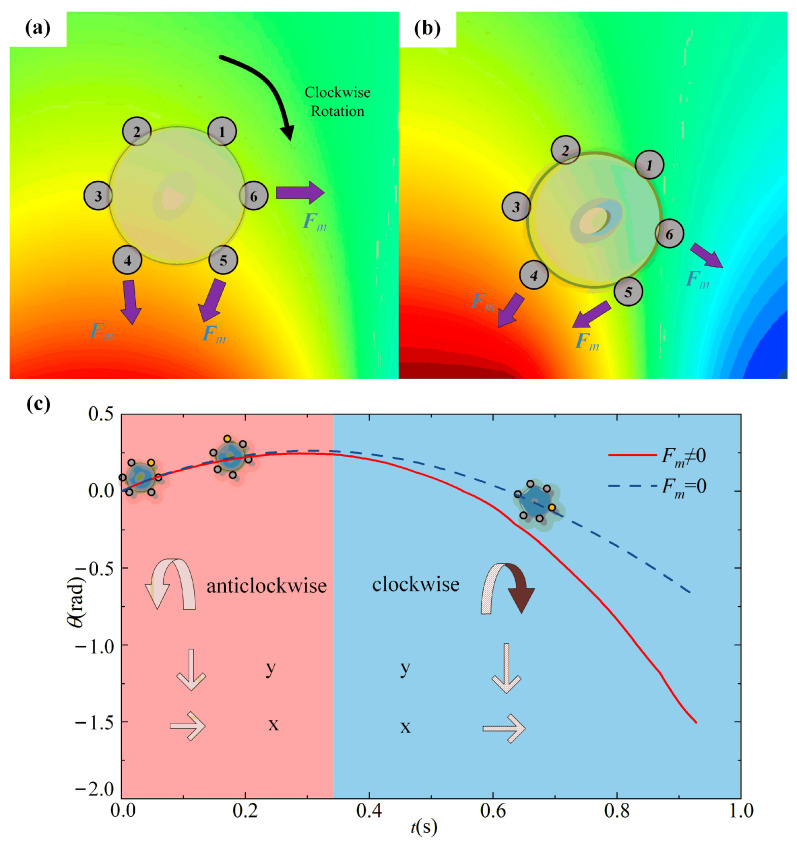
Analysis of rotation angle of tagged cells: (**a**) Schematic diagram of labeled cancer cells under stress; (**b**) Schematic diagram of labeled cancer cells after rotation; (**c**) tagged cell rotation angle curve.

**Figure 8 micromachines-14-02224-f008:**
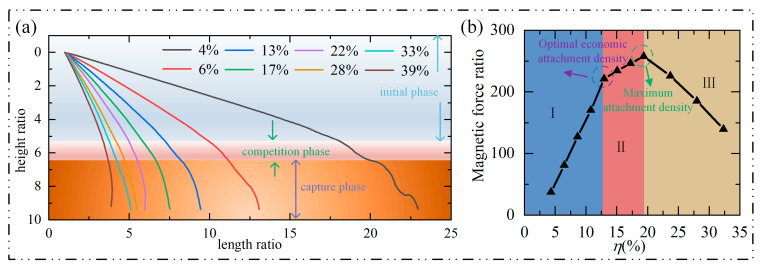
Motion analysis: (**a**) Labeled cell trajectories under different adhesion densities; (**b**) magnetic-force ratio at different adhesion densities.

**Figure 9 micromachines-14-02224-f009:**
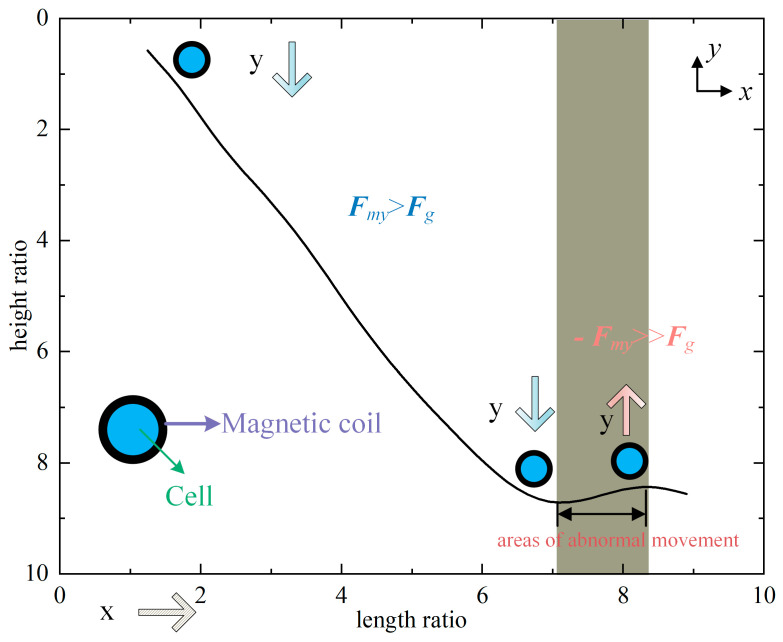
A partial motion path of a magnetic particle.

**Figure 10 micromachines-14-02224-f010:**
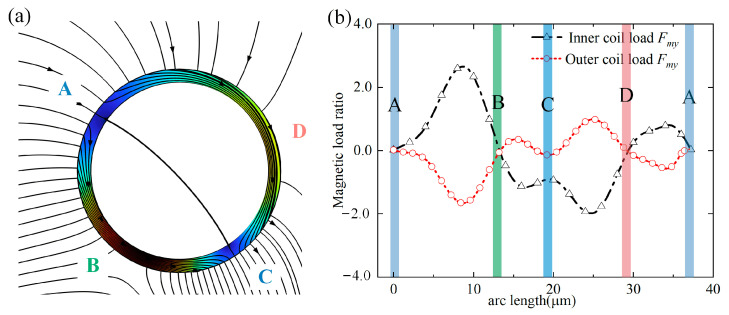
Force analysis in extreme conditions: (**a**) Schematic diagram of force on a ring of magnetic particles; (**b**) longitudinal magnetic-load curve of the inner and outer ring of a magnetic particle.

**Table 1 micromachines-14-02224-t001:** Calculate the domain geometry parameter table.

Calculation Parameters	Symbol	Value
Channel height	*H* (μm)	90
Magnetic block length	*l_1_* (μm)	20
Magnetic block height	*h* (μm)	5
Magnetic Block Pitch	*l_2_* (μm)	20
target radius	*R* (μm)	7
Magnetic particle radius	*r* (μm)	1

## Data Availability

This publication contains all the data gathered or analyzed during this investigation.

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
