# Peer review of "Numerical Studies on the Motions of Magnetically Tagged Cells Driven by a Micromagnetic Matrix"

_micromachines, 2023, doi:10.3390/mi14122224_

Round 1

Reviewer 1 Report

Comments and Suggestions for Authors

This manuscript aimed on building a two-dimensional simulation model for capturing magnetic beads from a magnetic matrix for separating magnetic particles from nonmagnetic fluids. It shows that the capture process can be divided into three stages: the initial phase, specific to the magnetic-lean region; the competition phase, which explains motion will be unpredictable due to uneven force; and the capture phase, which symbolizes the outcome of the competition phase. For the cell capture, two ideas of optimal and maximal attachment densities were presented by analyzing the capture trajectories of marked cells at different attachment densities. The results revealed that the Magnetic load at some points has an upward direction and is significantly stronger than gravity, which causes the target cells to travel upward and makes it challenging for the magnet matrix to catch them. The work has shown a qualitative description of the relationship between the capture trajectory and the micromagnetic matrix with solid results and give the experimental researchers some reference for the design of magnetic field and choose the proper magnetic particles for capture cells. However, there are some issues should be addressed for further discussion.

1.     In which cases, the gravity force can be negligible in Fig. 6? Are there some cases in the practical experiments to verify this assumption? please give an example.

2.     Is there any correlation between the particle size and the adhesion density? As the size increased the force for adhesion should change somewhat, the author may give some comments on it or show some results which should be important for the experimental investigation.

3.     In the flow condition, Reynold number is an essential parameter for the flow behavior. However, we could see few discussions on the fluidic conditions for the cell capture. How does it affect the force, the torque and the rotation of cells by changing the flow conditions?

4.     For cell research, the working temperature is normally 37 ºC, and is it probably to affect the magnetic properties of the particles or the fluidic viscosity. If so, please give some comments.

Comments on the Quality of English Language

Minor editing of English language is required.

Author Response

Response to Referee #1:

Thank you for your comments on our manuscript. Those comments are all valuable and helpful for revising and improving our paper and the essential guiding significance of our research. We have studied the comments carefully and made corrections, which we hope meet with approval. The leading corrections in the paper are as follows:

  1. In which cases, the gravity force can be negligible in Fig. 6? Are there some cases in the practical experiments to verify this assumption? please give an example.

Response: Thank you for your valuable comments. It is much larger than the surrounding liquid due to the high density of the magnetic particles; the magnetic particles sink into the fluid too quickly to allow for a thorough analysis of their motion trajectories under the influence of gravity, magnetism, and flow fields. During the movement, gravity remains constant. Here, the significance of magnetic and flow fields—rather than the effect of gravity—in drawing magnetic particles is emphasized. Bongseop et al. [1] suggested a magnetic force gradient-based microfluidic system called the Mag-Gradient Chip. They postulated that the two primary forces impacting the particles in this system were magnetic force and medium fluid drag. This magnetic gradient-based microfluidic device ignores gravity. Considering your insightful feedback, we have incorporated a background description in this regard for the updated manuscript. See Section 3.2 for details.

  1. Is there any correlation between the particle size and the adhesion density? As the size increased the force for adhesion should change somewhat, the author may give some comments on it or show some results which should be important for the experimental investigation.

            Response: Thank you for your valuable comments. There is a relationship between adhesion density and particle size. Immunolabeling was utilized by Wen et al. [2] to identify non-magnetic tumor cells. To do this, immunomagnetic nanospheres (IMNs) were supplemented with antibodies against the epithelial cell adhesion molecule (EpCAM). This article describes a rigid structure link-based tagged cell model that illustrates the adhesion of particles to a cell surface. The stiff construction approach prevents the magnetic particles attached to the cell surface from moving during the catching process. We may anticipate that the force of adhesion will rise along with particle size. Wen et al.'s study also noted that nanomagnetic particles outperform micron-sized particles in terms of binding capacity and capture efficiency. This suggests that the force needed for adhesion will decrease with decreasing size.

  1. In the flow condition, Reynold number is an essential parameter for the flow behavior. However, we could see few discussions on the fluidic conditions for the cell capture. How does it affect the force, the torque and the rotation of cells by changing the flow conditions?

Response: We appreciate you taking the time to carefully read our work throughout the evaluation process. The model described in this article employs a micromagnetic matrix and a low Reynolds coefficient to collect magnetic particles or tagged cells. Laminar flow occurs when incompressible fluids pass along channels slowly. The fluid's flow velocity is highest in the center of the channel and lowest near the wall. The viscous force and pressure differential the labeled cells experience in the fluid cause them to rotate. According to this study, the total of the pressure and viscous force is the fluid load (FT). The channel height, fluid density, and dynamic viscosity of the fluid are all fixed parameters in the model covered in this article. According to the definition of the Reynolds coefficient, changes in the average flow velocity in the channel might have an effect on the coefficient. When the Reynolds coefficient varies, so can the fluid load FT on the particles. Because of the rapid magnetic reaction rate of magnetic particles, this study focuses on the driving force of magnetic load rather than on fluid flow. To alter the current model, we will also take the fluid flow velocity into account. We have included a thorough discussion of fluid mechanics in Section 3.3.1 of the revised manuscript as a result of your insightful ideas.

  1. For cell research, the working temperature is normally 37 ºC, and is it probably to affect the magnetic properties of the particles or the fluidic viscosity. If so, please give some comments.

Response: We appreciate you taking the time to carefully read our work throughout the evaluation process. Particle sizes in the micron range are used in the model presented in this article. Under normal temperature circumstances, micron-sized magnetic particles can be used in a variety of applications. For the fluid viscosity factors in this article, Sutland's formula is utilized. The model presented in this article takes temperature effects on fluid viscosity into account.

[1] B Kwak, J Lee, D Lee, et al. Selective isolation of magnetic nanoparticle-mediated heterogeneity subpopulation of circulating tumor cells using magnetic gradient based microfluidic system. Biosensors and Bioelectronics, 2017, 88: 153-158. DOI: 10.1016/j.bios.2016.08.002.

[2] C Y Wen, L L Wu, Z L Zhang, et al. Quick-response magnetic nanospheres for rapid, efficient capture and sensitive detection of circulating tumor cells. ASC Nano, 2014, 8(1): 941-949. DOI :10.1021/nn405744f.

Reviewer 2 Report

Comments and Suggestions for Authors

The manuscript “Numerical studies on the motions of magnetically tagged cells driven by a micromagnetic matrix” by Tao et al. is a well-written study concerning the simulation of magnetic cell capture.

The references are balanced and the simulation parameters are described.

Introduction: While the different simulation models are well discussed and reviewed, there is only few statements of their actual applications. Thus can you also provide what kind of experimental systems have been simulated by the respective groups?

Moreover, there is no mentioning to which kind of particles and especially which kind of cells these models refer to. Can a bacterial cell separation predicted in a same manner as a cancer cell separation. This would be very helpful if you can provide literature.

Figure 1 references Equations 17 and 18 which I cannot find in the manuscript. Please provide the equations.

Which material do you consider for the magnetic particles with a density of 4 x 10 kg/m3 and relative magnetic permeability is set to 1, the magnetic permeability is set to 200.

Can you also explain in an equation what you mean with the relative and the magnetic permeability? I am a bit confused with the values used here.

If you use particles of 1 µm radius and target cells of 7 µm radius, I recommend not to use the word adsorption for the interaction of these systems since adsorption refers to molecular interaction and specifically the binding of molecules to a surface. Maybe adhesion is a better word or just use interaction.

You mention the cell rotation in the magnetic field. Can you cite previous studies predicting this effect? Moreover, can you also refer to the impact of this dynamic?

Line 505: The schematic illustration of the device shown in Figure 1 in [31] is consistent with the capture model provided in this article, which depicts the channel as having a bumpy structure.

This references figure 10 and not 1

Conclusion: This conclusion summarizes well the study but may you also give a brief outlook how these results might be applied?

Round 2

Reviewer 2 Report

Comments and Suggestions for Authors

The manuscript has been significantly improved.

There is only one minor comment about magnetic permeability is set to 200.

Can you please provide a reference for this value. Especially since you consider small superparamagnetic particles. Usually, the permeability is much lower from my experience (in the range of 3-10) for these particles.
